
# Measurements of traffic dominated pollutant emissions in a Chinese megacity.

Freya A. Squires[1], Eiko Nemitz[2], Ben Langford[2], Oliver Wild[3], Will S. Drysdale[1, 4], W. Joe F. Acton[3], Pingqing Fu[5, 6], C. Sue. B. Grimmond[7], Jacqueline F. Hamilton[1], C. Nick Hewitt[3], Michael Hollaway[3, a], Simone Kotthaus[7, b], James Lee[1, 4], Stefan Metzger[8, 9], Natchaya Pingintha-Durden[8], Marvin Shaw[1], Adam R. Vaughan[1], Xinming Wang[10], Ruili Wu[11], Qiang Zhang[11], and Yanli Zhang[10]

[1]Wolfson Atmospheric Chemistry Laboratories, Department of Chemistry, University of York, York, YO10 5DD
[2]Centre for Ecology and Hydrology, Edinburgh, EH26 0QB
[3]Lancaster Environment Centre, Lancaster University, Lancaster, LA1 4YQ
[4]National Centre for Atmospheric Science, University of York, York, UK
[5]Institute of Atmospheric Physics, Chinese Academy of Sciences, Beijing, China
[6]Institute of Surface-Earth System Science, Tianjin University, Tianjin, China
[7]Department of Meteorology, University of Reading, Reading, UK
[8]National Ecological Observatory Network Program, Battelle, 1685 38th Street, Boulder, CO 80301, USA
[9]University of Wisconsin-Madison, Department of Atmospheric and Oceanic Sciences, 1225 West Dayton Street, Madison, WI 53706, USA
[10]Guangzhou Institute of Geochemistry, Chinese Academy of Sciences, Guangzhou 510640, China
[11]Ministry of Education Key Laboratory for Earth System Modelling, Department of Earth System Science, Tsinghua University, Beijing, China
[a]Now at: Centre for Ecology & Hydrology, Lancaster Environment Centre, Bailrigg, Lancaster, UK
[b]Now at: Institut Pierre Simon Laplace, École Polytechnique, Palaiseau, France

**Correspondence:** James Lee (james.lee@york.ac.uk)

**Abstract.** Direct measurements of $NO_x$, CO and aromatic VOC (benzene, toluene, $C_2$-benzenes and $C_3$-benzenes) flux were made for a central area of Beijing using the eddy covariance technique. Measurements were made during two intensive field campaigns in central Beijing as part of the Air Pollution and Human Health (APHH) project, the first in November – December 2016 and the second during May – June 2017, to contrast winter and summertime emission rates. There was little difference in the magnitude of $NO_x$ flux between the two seasons (mean $NO_x$ flux was 4.41 mg m$^{-2}$ h$^{-1}$ in the winter compared to 3.55 mg m$^{-2}$ h$^{-1}$ in the summer). CO showed greater seasonal variation with mean CO flux in the winter campaign (34.7 mg m$^{-2}$ h$^{-1}$) being over twice that of the summer campaign (15.2 mg m$^{-2}$ h$^{-1}$). Larger emissions of aromatic VOCs in summer were attributed to increased evaporation due to higher temperatures. The largest fluxes in $NO_x$ and CO generally occurred during the morning and evening rush hour periods indicating a major traffic source with high midday emissions of CO indicating an additional influence from cooking fuel. Measured $NO_x$ and CO fluxes were then compared to the MEIC 2013 emissions inventory which was found to significantly overestimate emissions for this region, providing evidence that proxy-based emissions inventories have positive biases in urban centres. This first set of pollutant fluxes measured in Beijing provides an important benchmark of emissions from the city which can help to inform and evaluate current emissions inventories.





# 1 Introduction

Rapid development and population growth has led to an ever increasing number of "megacities", defined by the United Nations (UN) as a "metropolitan area with a total population of more than 10 million people" (United Nations' Department Of
Economic and Social Affairs: Population Division, 2016). In addition to being home to a large population, megacities are typically associated with high levels of industrialisation and extensive transportation networks making air pollution a common problem. Beijing is one such city that regularly experiences significant air quality problems. High levels of particulate matter (PM) in Beijing during winter months have been widely reported. In 2017, annual $PM_{2.5}$ (PM with a diameter less than 2.5 $\mu$m) concentrations reached 58 $\mu$g m$^{-3}$, approximately six times greater than the World Health Organisation (WHO) guideline
(Ministry of Ecology and Environment, the People's Republic of China, 2018). During the summer, concentrations of ozone, $O_3$, a major component of photochemical smog, regularly exceeded the WHO 8-hour mean limit of 100 $\mu$g m$^{-3}$ in Beijing. Both PM and $O_3$ have detrimental impacts on public health and both are formed in the atmosphere from reactions by precursor emissions that include nitrogen oxides ($NO_x$), carbon monoxide (CO) and volatile organic compounds (VOCs).

$NO_x$, the sum of nitrogen oxide (NO) and nitrogen dioxide ($NO_2$), and CO are two key anthropogenic pollutants. They are
damaging to human health in their own right as well as forming secondary pollutants, PM and $O_3$. China is the largest $NO_x$ emitter globally and is estimated to contribute as much as 18 % of global $NO_x$ emissions (European Database For Global Atmospheric Research, 2000-2012) while Beijing itself is reported to have annual mean $NO_2$ concentrations 16 $\mu$g m$^{-3}$ higher than the national average (Ministry of Ecology and Environment, the People's Republic of China, 2018). In high concentrations, $NO_2$ is a respiratory irritant (Strand et al., 1998; Tunnicliffe et al., 1994). CO is a harmful air pollutant produced from incom-
plete combustion processes including those used in power generation and from vehicle engines. Liu et al. (2018) concluded that there is an association between short-term exposure to ambient CO and increased cardiovascular disease mortality, especially coronary heart disease mortality. For both these pollutants, traffic emissions tend to be the dominant source in megacities.

In order to manage air quality it is vital that legislators have a clear understanding of pollutant emissions to guide abatement strategies. Models of atmospheric chemistry provide an important mechanism to predict the efficacy of abatement measures
on future air quality yet these predictions are only as certain as the emission inventories upon which they are based. For example, previous studies have highlighted large discrepancies between emissions inventories and measured emissions for UK cities both for $NO_x$ (Lee et al., 2015; Vaughan et al., 2016) and VOCs (Langford et al., 2010; Valach et al., 2015). Karl et al. (2018) revealed that non-methane VOC (NMVOC) emissions could be significantly higher than those used in most models by measuring emissions from Innsbruck, Austria. Inventories in China are associated with large uncertainties and are
rapidly changing in response to economic development and new environmental regulations. Saikawa et al. (2017) reviewed and compared five different emissions inventories for China and found large disagreements between them. Thus there is a critical need for reliable field measurements in order to further improve the emission estimates and reduce the uncertainty of inventories at local and regional scales (Zhao et al., 2017).





Given this pressing need for measurements of pollutant emissions, fluxes of $NO_x$, CO and commonly co-emitted VOCs (benzene, toluene, $C_2$-benzenes and $C_3$-benzenes) were calculated using the eddy-covariance (EC) technique for an urban area in Beijing. To the knowledge of the authors, this is the first time these emissions have been directly quantified in Beijing. This work was carried out as part of the Air Pollution and Human Health (APHH) Beijing project and an overview of this campaign
can be found in Shi et al. (2019).

## 2  Methodology

### 2.1  Site Description

Measurements were taken from an inlet part-way up a 325 m meteorological tower at the Institute of Atmospheric Physics, Chinese Academy of Sciences (IAP, CAS) (39°58'28"N, 116°22'16"E) in central Beijing. The site is between the third and
fourth ring roads and surrounding land use can be characterised as urban, being mainly residential with some busy (two and three lane dual-carriageway) roads nearby. The Jingzang Highway is approximately 400 m east of the site. Building heights surrounding the tower are predominantly 15 – 30 m in height, but with some almost 100 m tall within 500 m to the south of the tower. The site is in a 'green' area with some park space with a canal close by. Measurements were made over two field campaigns; the winter campaign from 05 November 2016 – 11 December 2016 and the summer campaign from 22 May 2017
– 25 June 2017 to allow a seasonal comparison of emissions.

Instrumentation was housed in a temporary shipping container laboratory located at the base of the tower. Sample lines from an inlet platform at an elevation of 102 m ran down the tower to the laboratory. Air for sampling was drawn down a $\frac{1}{2}$" O.D. (I.D. 9 mm) perfluoroalkoxy (PFA) tube at a rate of approximately 95 L min$^{-1}$ resulting in an inlet pressure of 44 kPa. This ensured turbulent flow was maintained (Reynolds Number $\approx$ 7000) and attenuation of signals along the $\sim$ 120 m sample
line were minimised. Particles were removed from the airflow via a 90 mm Teflon filter mounted near the inlet which was changed at 24 hour intervals. The inlet of the tube comprised of a custom built, 32 mm diameter, stainless steel manifold cap with gauze to prevent larger debris entering the tube. The manifold was mounted 82 cm vertically below a sonic anemometer (Model HS-50, Gill Instruments) which measured the three wind vectors, u, v and w data at a rate of 10 Hz. During the winter campaign it was orientated NW from the tower and during the summer campaign towards the SE to measure the main wind
direction without obstructions. However, analysis of the turbulence characteristics did not suggest that the open structure of the tower affected the measurements even when the flow came through the tower. This may be due to the size of the eddy-motions at this measurement height.

### 2.2  Instrumental Description

#### 2.2.1  $NO_x$ Sampling and Measurement

Concentrations of $NO_x$ were measured using a dual-channel chemiluminescence instrument (Air Quality Designs Inc., Colorado). The instrument is similar to that described in Lee et al. (2009), but modified to enable high time resolution data to be





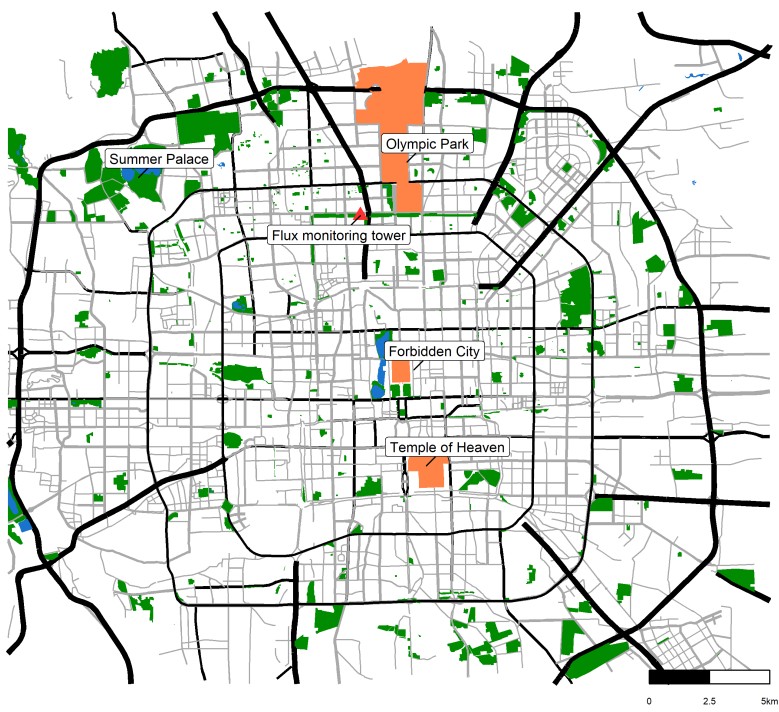

**Figure 1.** Measurement site position is shown by the red triangle between the third and fourth ring roads. Key landmarks of Beijing are highlighted in orange with major roads shown in black and smaller roads in grey. Parks are shown in green and water in blue. Surrounding land use is mainly residential with many restaurants within a few hundred meters of the site with the Jingzang Highway close by. Map was built using data from © OpenStreetMap contributors 2019. Distributed under a Creative Commons BY-SA License.

collected with a residence time of 0.12 s inside the photolytic conversion cell. NO was measured directly by chemiluminescence from the reaction of NO and $O_3$ in one channel. The second channel measures total $NO_x$ via photolytic conversion of $NO_2$ to NO, at a wavelength of 395 nm, and then by chemiluminescence reaction with $O_3$, as per the direct measurement of NO. Instrument data were recorded at a frequency of 5 Hz.

5      The $NO_x$ instrument was calibrated regularly (every 2 – 3 days) throughout the campaign using NO gas standards traceable to the UK's National Physical Laboratory's (NPL) NO scale. The instrument was calibrated via standard addition of a small flow of NO calibration gas to a flow of $NO_x$-free ambient air ($NO_x$ was removed using a Sofnofil/charcoal trap). The sensitivities of the NO and $NO_x$ channels were calculated by direct addition of the diluted NO calibration gas. The $NO_2$ – NO conversion efficiency within the $NO_x$ channel was calculated by gas-phase titration of the diluted NO calibration gas with $O_3$ to create a

10    known quantity of $NO_2$. During the calibration cycle, an instrument zero was quantified by diversion of sample flow to 'zero volumes' so that the chemiluminescence reaction was completed before the gas reached the detectors. Zero measurements were scheduled to occur for 15 seconds every hour through the normal operating schedule.





### 2.2.2 CO Sampling and Measurement

CO was measured using a resonance fluorescent instrument (Model AL5002, Aerolaser GmbH, Germany). Flows were adjusted to reduce cell lag times so data could be recorded at 5 Hz to match the $NO_x$ data acquisition rate. Details of the unmodified system are described by (Gerbig et al., 1996, 1999). The CO instrument was calibrated regularly (every 2 – 3 days) throughout
the campaign as for the $NO_x$ instrument using a 1 ppm CO in synthetic air standard. Previous urban flux measurements with this type of instrumentation have been presented for UK cities by (Famulari et al., 2010; Harrison et al., 2012; Helfter et al., 2016).

### 2.3 VOC Sampling and Measurement

VOCs were measured using a Proton Transfer Reaction Mass Spectrometer (PTR-MS). The PTR-MS (PTR-MS 2000, Ionicon
Analytik, Innsbruck, Austria) was installed at the base of the tower and subsampled from the common inlet line at 30 sccm. The instrument was operated with a 5 Hz measurement frequency. The drift tube maintained at 60 °C, with a pressure of 1.9 mbar and 490 V applied across it. This gave an E/N (the ratio between electric field strength and buffer gas density) of 120 Td in the drift tube. This set up is described in more detail by Acton et al. (*in prep.*).

The PTR-MS was calibrated twice a week during both the winter and summer campaigns using a VOC standard containing
methanol, acetonitrile, ethanol, 1,3-butadiene, acetone, isoprene, butenone, butan-2-one, benzene, toluene, m-xylene and 1,2,4-trimethylbenzene at 1 ppmv (National Physics Laboratory, Teddington, UK). The standard was dynamically diluted in zero air to provide a six point calibration. In the winter campaign the instrument also was calibrated using two Ionicon standards the first containing methanol, acetonitrile, acetaldehyde, ethanol, acrolein, acetone, isoprene, crotonaldehyde, butan-2-one, benzene, toluene, o-xylene, chlorobenzene, $\alpha$-pinene and 1,2-dichlorobenzene at 1 ppmv each and the second made up of
formaldehyde, acetaldehyde, acrolein, propanal, crotonaldehyde, butanal, pentanal, hexanal, heptanal and octanal at 1 ppmv, nonanal at 600 ppbv and decanal at 500 ppbv. PTR-MS data was processed using PTRViewer (Ionicon Analytik).

### 2.4 Data Processing

Prior to the calculation of pollutant fluxes, the raw data were scaled to take account of calibrations. The sensitivity within each channel of the $NO_x$ chemiluminescence instrument remained consistent throughout the winter and summer campaigns so
data were scaled using median sensitivity values. The conversion efficiency of the $NO_2$ channel gradually deteriorated over the two campaigns and so $NO_2$ data were scaled using linearly interpolated conversion efficiency values. For CO concentration data, the instrument sensitivity following each calibration was directly applied and the sensitivity remained consistent for the duration of the two campaigns.

Concentration data were coupled with wind data reported by the sonic anemometer by sub-sampling the wind vector data
to match the 5 Hz concentration data. Data were then despiked prior to flux calculation as per the method described in Brock (1986) and Starkenburg et al. (2016). Following despiking, the lag time between vertical wind velocity measured in-situ on the tower and the pollutant concentrations, measured on the ground, was calculated. The lag time correction applied was





determined by first allowing the software to calculate the optimum lag for each averaging period by maximisation of the cross-covariance between concentration and vertical wind component. Because there was no discernible pattern or trend in the lag-times and to prevent the flux bias that cross-covariance maximisation can introduce when fluxes are small (Langford et al., 2015), the final fluxes were calculated by applying the median time-lag value for each campaign to all flux periods. The lag

time between the concentration and vertical wind speed during the winter campaign was found to be 9.6 s, 10.0 s, 10.2 s for NO, $NO_2$ and CO respectively. For the summer campaign lag times were calculated as 9.4 s, 9.8 s and 10.6 s. Lag time correction was performed using the same method for the PTR-MS VOC concentrations. Lag times calculated for isoprene (summer data) and benzene (winter data) within a 5 – 15 s window and these values were then applied to all compounds. Where the lag time was found to be outside of the 5 – 15 s range a standard lag time of 9 s was applied.

## 2.5   Flux Calculations

The flux, $F$, of each species, which can be defined as the vertical transport of a pollutant per unit area per unit time, was then calculated using the eddy covariance (EC) method (Lee et al., 2005):

$$F \approx \overline{w'c'} \tag{1}$$

where $w'$ is instantaneous change in vertical wind speed (i.e. $w' = w - \bar{w}$, where overbars denote averages) and $c'$ is instan-

taneous change in pollutant concentration. The flux was calculated over a 30 minute averaging period and quantified using the eddy4R family of R-packages (Metzger et al., 2017) with a customized eddy-covariance workflow template to suit the requirements of our study. Random uncertainty was calculated using the method outlined by Mann and Lenschow (1994). The flux limit of detection was taken to be twice the random error. It should be noted that, due to the high measurement height, 30 minute fluxes might be an underestimation of the 'true' flux as the averaging period may not capture low frequency contribu-

tions. To quantify the effect, a comparison between 30 minute, 60 minute and 120 minute averaging intervals was carried out for a week-long period of the summer campaign which indicated 30 minute fluxes were 93 % of the 60 minute fluxes whilst 120 minutes fluxes were considered too long an averaging periods for sufficient temporal resolution. Additionally increasing the length of averaging time introduces more non-stationary periods into the data. The 30 minute flux is therefore a compromise between capturing the entirety of the flux by keeping the low frequency flux loss small (7 %) and having sufficient temporal

resolution to relate the measurements to real-world processes. Fluctuations in temperature and humidity can impact fluxes by causing variation in air density (Webb et al., 1980). For closed path systems, such as those used in this study, air density varia-tions caused by sensible-heat flux are negligible however variations due to latent heat flux may need to be corrected for. For CO fluxes, samples were passed through a dryer negating the need for this correction, however latent heat flux could have an impact on the $NO_x$ fluxes (Moravek et al., 2019). The magnitude of the correction is proportional to the concentration/flux ratio which

for reactive species, like $NO_x$, is small. The effect of latent heat flux on $NO_x$ fluxes was found to be significantly less than 1 % throughout the campaigns and so the WPL correction was not applied (Pattey et al., 1992). The effect of high-frequency spectral loss on $NO_x$ and CO fluxes was investigated using a wavelet-based methodology (Nordbo and Katul, 2013). Spectral losses were found to be less than 3 % and so were not corrected for.



### 2.5.1  Corrections and Filtering

There are numerous assumptions made when calculating EC fluxes, all of which can introduce uncertainties in the derived quantity. Further conditions need to be met for the measured flux to be representative of surface flux. Assumptions include but are not limited to, the flux being fully turbulent with all transport done by eddy transfer, the terrain being homogeneous, measurements being made within the boundary layer, air density fluctuations being negligible and conditions remaining stationary. A common method to deal with periods of low turbulence, during which the flux at the measurement height may not reflect the surface flux, is to filter the data based on a friction velocity ($u_*$) threshold. Friction velocity accounts for shear stress in the turbulent boundary layer, and can be calculated from the instantaneous wind vectors $u'$, $v'$ and $w'$ (Foken and Napo, 2017). Concepts for $u_*$ filtering were originally developed by the community measuring $CO_2$ exchange with vegetation. Here, incorrect application of $u_*$ filtering can lead to a "double-counting" of flux as described in Aubinet (2008). In addition, by filtering out low-turbulence cases (low $u_*$ values) the data set can become biased with little information about nighttime and winter periods. Whilst for $CO_2$ exchange with vegetation fairly robust parametrisations exist that can be used to gap-fill periods of low turbulence, no such information is yet available for urban fluxes. Liu et al. (2012) therefore argue against applying $u_*$ filtering for the IAP site during a similar analysis of $CO_2$ fluxes, and suggest more errors could be introduced through filtering than not. Thus $u_*$ filtering was also not applied to the data presented here, unless otherwise stated. Approximately 29 % of winter fluxes and 11 % of summer fluxes are associated with $u_*$ values below 0.175 m s$^{-1}$. Average fluxes as a function of $u_*$ values are presented in fig. A1 and the effect of $u_*$ filtering on diurnal variation is shown in fig. A2. These show the maximum possible effect of low turbulence on fluxes. Because low turbulence is correlated with nighttime conditions during which emission activity is reduced, an increasingly stringent $u_*$ filter preferentially removes periods during which the surface flux is smaller than the average. This may result in an increase in the average nighttime flux that does not necessarily reflect suppression of the flux by lack of turbulence.

Stationarity is another important consideration for flux data. Stationarity is when the flux is statistically invariant over the averaging period and is quantified using the method described in Foken and Wichura (1996). The stationarity criterion is likely not to be met when fluxes are small and subject to a large random uncertainty; this is irrespective of whether the conditions are actually non-stationary. As a result this filter tends to remove the smallest fluxes and can bias flux results (e.g. Nemitz et al., 2018). A broad stationarity filter of 60 % was applied to all flux data presented in any average diurnals, though non-stationary data is presented and highlighted in time series plots. This stationarity filter was used as a more rigorous filter of 30 %, commonly used within the $CO_2$ flux community, removed a large proportion of the data. During the winter campaign 23 % of NO$_x$ fluxes and 22 % of CO fluxes were non-stationary under this more rigorous criterion. During the summer campaign these proportions were 16 % of NO$_x$ fluxes and 39 % of CO fluxes. The 60 % threshold used was determined to be appropriate as it falls within the stationarity range recommended for "general use", such as using diurnal averages to interpret trends (Foken et al., 2004). Further to these corrections, any periods where the boundary layer was within 30 m of the measurement height were removed from the data. Boundary layer height was measured throughout both campaigns using a celiometer (Vaisala CL31).





It was also important to consider storage effects; due to the build-up or dilution of the pollutants below the measurement height. Build-up can occur during periods of low turbulence e.g. during the night and this accumulation reduces the flux at the measurement height with respect to the emission at the ground (Finnigan, 2006). As conditions become more turbulent the accumulated pollutant concentrations gets diuluted again and the measured flux contains a component that originates from the stored material rather than emission. Gas concentration profile measurements can be used to allow detection of build-ups by providing data for computing a storage term below measurement height. In this case, the storage flux, $F_s$, at time, $t$, was calculated according to the following equation (Andreae and Schimel, 1990):

$$F_{s(t)} = \frac{C_{(t - \frac{t}{2})} - C_{(t - \frac{t}{2})}}{t} \tag{2}$$

By comparing the evolution of the concentration at the single measurement height with that measured at three heights up the tower tower for $CO_2$, it was determined that calculating storage flux using the single concentration at the measurement height was a reasonable approximation of the storage within the column. It should be noted that this storage correction to some extent takes care of the flux suppression at low turbulence, except for the interaction with advection and chemistry.

Throughout, this paper focusses on the analysis of total $NO_x$ flux rather than NO and $NO_2$ flux separately. Whilst the two compounds undergo rapid interconversion the total should be conserved at the time-scale that governs the transport from the surface to the measurement height; the major loss route for $NO_x$ is $HNO_3$ formation through reaction with the hydroxyl (OH) radical. This loss is assumed to be negligible between ground emission and sampling at the tower inlet. Calculation of Deardorff velocity suggests that on average the time taken for a parcel of air to reach the 102 m measurement point is ~68 s (Deardorff, 1970). Assuming average OH concentrations of $1 \times 10^6$ molecules $cm^{-3}$ it is estimated that less than 1 % reacts with OH at this time-scale.

## 2.6 Footprint Model

A flux footprint is the area surrounding the measurement tower that contributes to a measured flux based on factors such as wind speed, wind direction, atmospheric stability and surface roughness. A statistical flux footprint model can be used to quantify the flux contribution of each cell of an emission grid relative to the distance away from the measurement position in all directions, creating a weighing matrix that estimates the ground influence of a particular cell contributing to the observed emission flux. A footprint was calculated for each half hour flux period at 100 m resolution. The footprint model used is described in detail in Kljun et al. (2004) and Metzger et al. (2012). Surface roughness values were taken from Liu et al. (2012) and taken to be 2.5 m, 3.0 m, 5.3 m and 2.8 m for the NE, SE, SW, and NW wind quadrants respectively. Figure 2 shows the average footprint for the winter and summer campaigns with the 30 %, 60 % and 90 % cumulative contributions to the measured flux represented by the contours.

For both campaigns the 90 % contribution to measured flux extended as far as 7 km from the measurement site for some averaging periods, however, as shown in fig. 2 on average 90 % of the contribution to measured emissions was within 2 km of the tower. Figure 2 shows the difference in areas of influence covered during the winter and summer campaigns due to differences in dominant wind directions. During the winter, the measured fluxes were predominantly from the north-west





encompassing Beitucheng West Road and a block of predominantly commercial buildings and restaurants. The mean footprint maxima, the distance away from the tower at which the maximum contribution to measured flux occurs, falls 0.26 km away from the tower in winter. In summer, the fluxes were mostly influenced by areas to the north east and east of the tower, encompassing the Jingzang Expressway. The mean footprint maxima for summer was also 0.26 km away from the site.

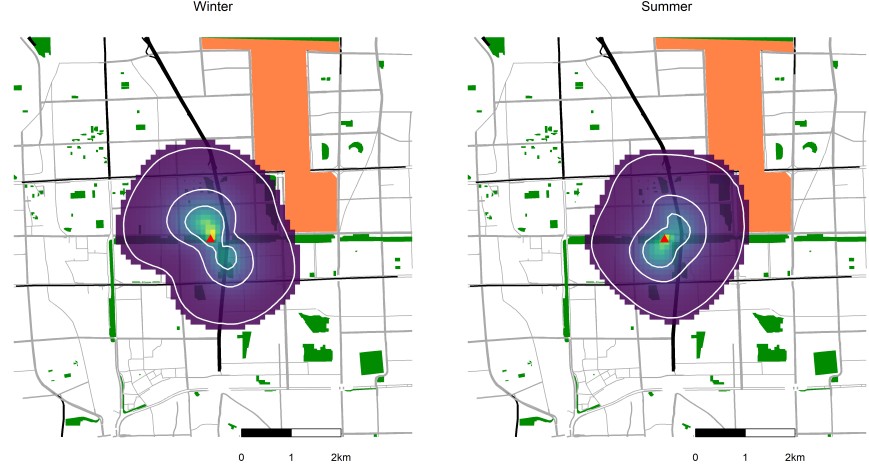

**Figure 2.** The mean flux footprints for the winter and summer campaigns. The site is shown by the red triangle in the centre of the map. Each square is 100 m$^2$ and the brighter colours indicate a greater influence on measured emission for a particular area. The white rings show the areas contributing 30 %, 60 % and 90 % to the flux total with inner ring representing the 30 % contribution and outer ring representing 90 % contribution. The 90 % of the influence from the footprint extends up to 2 km away from the tower with maximum contribution 0.26 km away from the tower. Map was built using data from © OpenStreetMap contributors 2019. Distributed under a Creative Commons BY-SA License.

## 2.7 Inventory

The measured pollutant fluxes were compared with the Multi-resolution Emissions Inventory for China (MEIC, Qi et al. (2017); http://www.meicmodel.org/) to evaluate how well the inventory describes the diurnal evolution of pollutants and their absolute magnitude. MEIC considers five emission source sectors; power plants, industry, transport, residential, and agricultural and is presented at a resolution of 3 km$^2$. Agricultural emissions are not relevant within our flux footprint so not included in this work. The NO$_x$ and CO emission predicted by the inventory are calculated from the multiplication of each footprint matrix by the MEIC grid. For this evaluation, an optimised version of the MEIC v1.3 inventory for 2013 was used that was derived by fitting the NAQPMS model with observed pollutant concentrations during the campaign periods (Du et al., 2019). Temporal variability is represented in the inventory with twelve emissions grids, one for each calendar month. The emissions throughout the month are assumed to be the same each day with a diurnal cycle imposed. Example emissions grid for Beijing are presented in figure A3 for November.





**Table 1.** Summary table for $NO_x$ and CO fluxes and concentrations. Data presented is for fluxes which are within 60 % stationarity criteria for all $u_*$ values.

| | Winter | | Summer | |
|---|---|---|---|---|
| **Concentration (mg m$^{-3}$)** | **NO$_x$** | **CO** | **NO$_x$** | **CO** |
| Mean | 0.103 | 1.41 | 0.0310 | 0.502 |
| Median | 0.0839 | 1.01 | 0.0213 | 0.429 |
| Percentiles | | | | |
|   5th | 0.0154 | 0.268 | 0.00810 | 0.210 |
|   95th | 0.252 | 3.60 | 0.0876 | 0.998 |
| Standard Deviation | 0.0776 | 1.16 | 0.0278 | 0.267 |
| **Flux (mg m$^{-2}$ h$^{-1}$)** | | | | |
| Mean | 4.41 | 34.7 | 3.55 | 15.2 |
| Median | 4.14 | 32.0 | 2.45 | 12.4 |
| Percentiles | | | | |
|   5th | -1.24 | -27.0 | -0.0139 | -2.15 |
|   95th | 10.6 | 103 | 11.5 | 42.9 |
| Standard Deviation | 3.86 | 40.1 | 3.69 | 14.4 |

## 3 Results & Discussion

Statistics for $NO_x$ and CO fluxes and concentrations presented in this work are shown in table 1. Figure 3 shows a time series of measured $NO_x$ and CO fluxes during the winter and summer measurement campaigns where grey coloured traces highlight data which does not meet the 60 % stationarity criteria. For winter, 3.5 % of $NO_x$ fluxes and 8.2 % of CO fluxes were non-stationary. In summer, slightly fewer measurements did not meet the stationarity criteria with 3.1 % of $NO_x$ fluxes and 7.3 % of CO fluxes falling outside the 60 % stationarity limits. The mean flux for $NO_x$ during the winter measurement period was $4.4 \pm 3.9$ mg m$^{-2}$ h$^{-1}$ and $3.6 \pm 3.7$ mg m$^{-2}$ h$^{-1}$ for the summer measurement period. For CO, there was a larger difference between the two seasons with the mean flux calculated as $35 \pm 40$ mg m$^{-2}$ h$^{-1}$ for winter and $15 \pm 14$ mg m$^{-2}$ h$^{-1}$ in summer. Some of the calculated fluxes are negative, corresponding to deposition to the surface, however, as expected in an urban environment, the net flux is strongly positive indicating emission. The average $NO_x$ fluxes for the winter and summer periods are similar suggesting an emission source that does not change much between seasons. In contrast the average CO flux is over double in the winter compared to the summer indicating an additional source in the winter.

When considering previous literature, the $NO_x$ fluxes measured in Beijing were low compared to London, UK where net emissions were in the range of 10.8 mg m$^{-2}$ h$^{-1}$ – 14.4 mg m$^{-2}$ h$^{-1}$ (Lee et al., 2015). A study investigating $NO_x$ fluxes across 13 urban locations in Norfolk, Virginia reported values in the range of 18 – 28 mg m$^{-2}$ h$^{-1}$, up to 8 times higher than those measured in Beijing (Marr et al., 2013). Fluxes measured in Beijing were similar to those measured in Innsbruck at a roadside site in July – October, 2015 where $NO_x$ fluxes of 2.5 – 5.2 mg m$^{-2}$ h$^{-1}$ were reported (Karl et al., 2017). For CO, measured





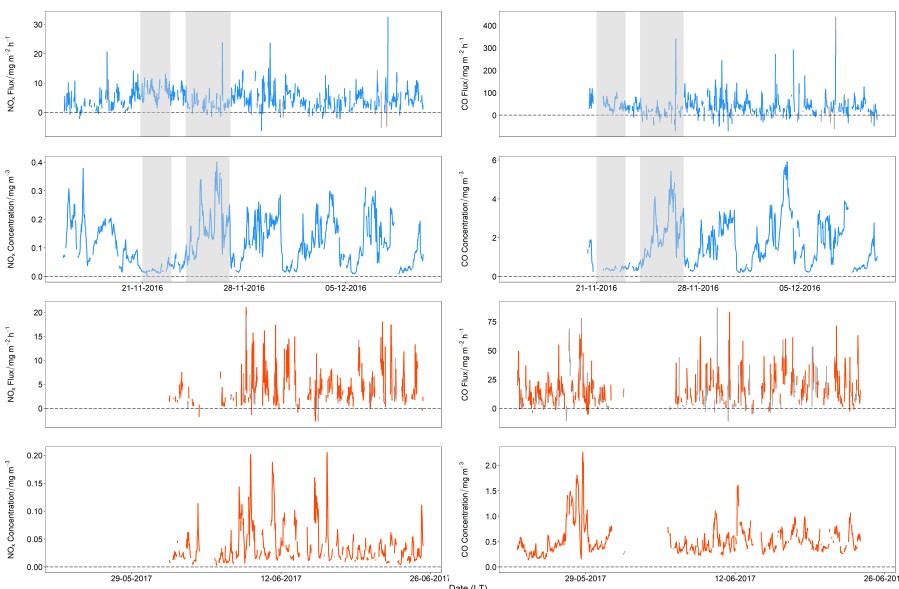

**Figure 3.** Time series data for 30 minute averaged $NO_x$ and CO fluxes for the summer and winter campaign; the blue trace corresponds to measurements taken during the winter campaign and the orange the summer campaign. Grey coloured flux traces correspond to fluxes outside of the 60 % stationarity criteria. Gaps in the time series are due to instrument problems. The grey boxes on the winter timeseries plots highlight two contrasting periods discussed in section 3.2.

fluxes in central London were 2 – 3 times lower than those measured in Beijing. Average winter (December – February) CO flux was reported to be $12.5 \pm 3.4$ mg m$^{-2}$ h$^{-1}$ and average summer (June – July) CO flux was reported to be $4.0 \pm 0.1$ mg m$^{-2}$ h$^{-1}$, for the measurement period September 2011 – December 2014 (Helfter et al., 2016). $NO_x$ emissions are likely to be lower in Beijing than for other cities as the majority of onroad vehicles in Beijing are light-duty gasoline vehicles (LDGVs)

which made up to 93 % of the vehicle fleet in Beijing in 2013 (Yang et al., 2015) compared to other cities which have a higher proportion of diesel vehicles. However, because fluxes vary spatially within each city, care needs to be taken when comparing measurement datasets as the type of the location within the city needs to be considered.

### 3.1   Average diurnal cycles

Figure 4 shows the mean diurnal profile for both campaigns for pollutant fluxes, concentrations and mixing layer height. Diurnal
profiles are a useful way to visualise flux data, as time of day may indicate processes responsible for emissions. During the winter campaign, the $NO_x$ fluxes were lower during the early hours of the morning (between 00:00 – 05:00) ranging between 1.9 mg m$^{-2}$ h$^{-1}$ and 3.6 mg m$^{-2}$ h$^{-1}$. After 06:00, the $NO_x$ fluxes increased and remained elevated, though variable, throughout the day with a mean daytime value (06:00 – 18:00) of 5.1 mg m$^{-2}$ h$^{-1}$. The $NO_x$ fluxes decreased again in the evening. The daily variability in CO fluxes followed a similar pattern to $NO_x$. The mean daytime CO flux was 38 mg m$^{-2}$ h$^{-1}$, and was lower during
the night (19:00 – 05:00) with a mean nighttime value of 29 mg m$^{-2}$ h$^{-1}$. Concentrations are influenced by meteorology and





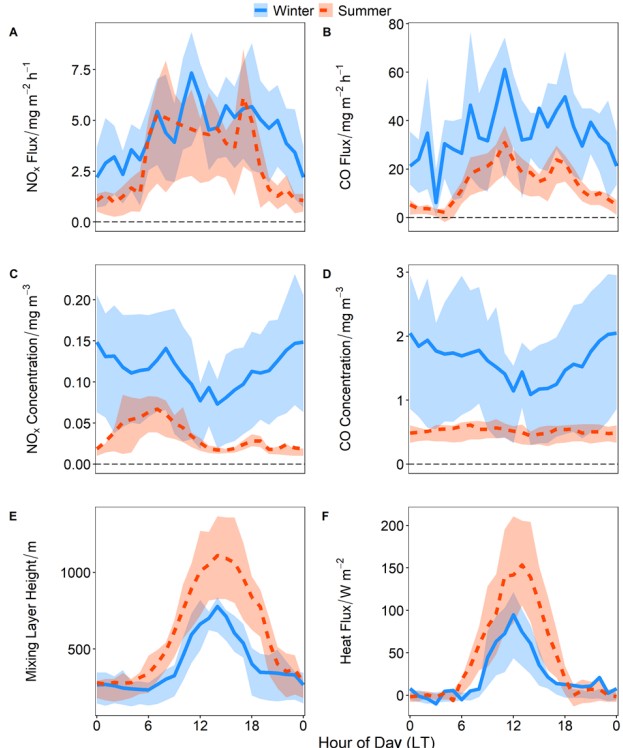

**Figure 4.** Average diurnal profiles for A) $NO_x$ flux, B) CO flux, C) $NO_x$ concentration, D) CO concentration and E) mixing layer height and F) heat flux. Blue, solid lines are for measurements taken during the winter campaign and the orange, dashed lines are measurements taken during the summer campaign. The shaded areas represent the 25th and 75th percentiles to show the spread of the data. For the $NO_x$ and CO fluxes, only stationary data has been used but no $u_*$ filtering has been applied when doing the diurnal averaging.

long range transport as well as local emissions (in this study local emissions refer to emissions from within the flux footprint). $NO_x$ and CO concentrations remained constant during the night due to stability in the mixing layer height. When $NO_x$ and CO emissions increased after 05:00, this enhancement in flux was reflected in the concentration data with a small peak in $NO_x$ and CO concentrations around 06:00, though the effect is masked as the mixing layer height begins to increase. Concentrations

5  decreased to their minima at 15:00 when the mixing layer height reached its highest point and increase again when the mixing layer height contracted over night.

For the summer campaign, fluxes are slightly lower than during the winter. The mean nighttime flux was less than 1.6 mg m$^{-2}$ h$^{-1}$ for $NO_x$ and 6.5 mg m$^{-2}$ h$^{-1}$ for CO. Emissions rapidly increased after 05:00 and started to decrease at around 17:00. Daytime emissions for $NO_x$ were fairly consistent with a mean value of 4.6 mg m$^{-2}$ h$^{-1}$. The first of two distinct peaks in

10  the observed $NO_x$ fluxes occurred at 07:00, where emissions reached 5.4 mg m$^{-2}$ h$^{-1}$ and the second occurred at 17:00 with a slightly higher value of 6.3 mg m$^{-2}$ h$^{-1}$. The daytime profile for CO showed two distinct peaks in emissions; the first peak occurred at 11:00 with CO emissions around 30 mg m$^{-2}$ h$^{-1}$ and the second at 17:00, when CO fluxes were around 24 mg m$^{-2}$




h$^{-1}$. Between these times the CO emissions dipped with a minimum daytime value of 16 mg m$^{-2}$ h$^{-1}$ at 14:00 – 15:00. The influence of local emissions on concentration is more clearly observed during the summer campaign; the peaks in NO$_x$ and CO fluxes at 17:00 occurred at the same time as an enhancement in NO$_x$ and CO concentration. Both NO$_x$ and CO concentrations reached their daytime minima at 14:00 - 15:00 when the mixing layer height was at its peak.

Given that the NO$_x$ emissions were fairly consistent between the two seasons it is likely that the major sources of NO$_x$ do not vary significantly over the year. In urban areas, vehicular emissions tend to be a dominant source of NO$_x$ (Parrish et al., 2009; von Schneidemesser et al., 2010; Borbon et al., 2013) and previous studies measuring NO$_x$ fluxes have attributed emissions to vehicles (Lee et al., 2015; Vaughan et al., 2016). Diurnal variation in summer NO$_x$ emissions agree well with previously reported diurnal variation in Beijing's traffic flow. Jing et al. (2016) show that traffic flow (vehicle number per hour) begins to

increase from 05:00 in the morning, corresponding to the observed increase in NO$_x$ emissions. As would be expected, peak traffic flow coincided with lowest average vehicle speeds and occurred at 08:00 and 18:00. NO$_x$ emissions are dependent on fuel type, engine type, combustion temperature, vehicle speed, engine load and exhaust after-treatment technology. It is known that "stop - start" driving conditions and idling can enhance NO$_x$ emissions compared to driving at steady speeds. Observations indicated that there are peaks in NO$_x$ emissions during these rush hour periods. CO emissions for summer also suggested a

strong traffic influence, although the peak at 11:00 may indicate an additional source, possibly relating to cooking given the time of day. Cooking was identified as a major contributor to the organic PM$_1$ flux in aerosol flux measurements made during the same measurement period (Langford et al. *in prep.*). NO$_x$ and CO fluxes measured during the winter also indicated some traffic influence, although the trend is less clearly resolved than for the summer measurements. Winter NO$_x$ emissions had a peak of 6.0 mg m$^{-2}$ h$^{-1}$ at 07:00 and there was a broader evening peak of similar magnitude between 17:00 – 18:00. NO$_x$ flux

peaked at 11:00 at 6.7 mg m$^{-2}$ h$^{-1}$. CO flux was more variable; there was a small peak around the time of the morning rush hour though this was not as clearly resolved as it is for NO$_x$. CO flux peaked at 11:00 and in the evening though the profile showed greater short term variability for NO$_x$.

    The difference between the winter and summer diurnal averages was more significant for CO than for NO$_x$ with larger CO emissions in the winter than the summer throughout the day. This may be in part due to an additional source unique to winter,

for example domestic heating. Local heating sources are likely to be reasonably consistent throughout the day which may go some way to explaining why the relative difference between nighttime and daytime emissions is smaller than for NO$_x$. Given the similarity between the NO$_x$ and CO diurnal variability, there is likely to be a traffic influence for CO. Strong seasonal variability in the fluxes of CO has been observed for London as another megacity where CO emissions measured in summer were 69 % lower than in winter (Helfter et al., 2016). In this case, higher winter CO emissions were attributed mainly to

vehicle cold starts and reduced fuel combustion efficiency due to colder ambient temperatures. It should be noted that this study took place in a temperate, developed city and Beijing needs more winter heating which until recently, was primarily from coal. Indeed, Langford et al. (*in prep.*) identified signatures of coal and solid fuel combustion within the flux footprint of the measurement although residential heating is overwhelmingly dominated by distance heating in this area of Beijing.

    Beijing has attempted to reduce emissions through traffic management as well as by imposing emissions reduction regula-

tions; congestion reduces vehicle speed which can increase emissions (Yang et al., 2015). One management strategy imposed





restrictions on heavy duty vehicles (HDVs) entering the city (past the sixth ring road) and only permits non-local vehicles to enter between 00:00 – 06:00. HDVs, particularly those using diesel fuel, are thought to be responsible for 85 % of $NO_x$ emissions, whilst light duty vehicles (LDVs) are considered responsible for more than half of CO emissions (Yang et al., 2015). Nighttime emissions of $NO_x$ and CO could be attributed to HDVs making up an unusually high proportion of the vehicle fleet.

Yellow label vehicles, vehicles which do not meet the China I emissions standard have been forbidden from entering Beijing since 2014.

## 3.2    Impact of local emissions on air quality

The average diurnal profiles in fig. 4 highlight the relationship between emissions, mixing layer height and concentrations. Whilst emissions and concentrations seem to be closely linked when averaged over the whole campaigns, there are periods

where local emissions do not drive concentrations. During the winter months Beijing experiences a frequent cycling between "polluted" and "clean" days and this phenomenon has been termed "sawtooth cycles". During winter pollutants build up during near stagnant periods with SE wind flow being trapped by the mountain range in the NW. These are then advected out of the city when wind speed increases and the direction switches to the NW resulting in sharp reductions in atmospheric concentrations (Jia et al., 2008; Li et al., 2017). This distinctive meteorological phenomenon occurs in Beijing as a result of the East Asian

Winter Monsoon, itself driven by temperature differences between the Pacific Ocean and Asian continent (Chen et al., 1992). This cycling was observed during the winter field campaign and can be seen in $NO_x$ and CO concentrations highlighted in fig. 3. During the period 21 November 2016 – 23 November 2016 average daytime concentrations of 0.020 mg m$^{-3}$ and 0.35 mg m$^{-3}$ for $NO_x$ and CO respectively were observed. On the following three days, 24 November 2016 – 27 November 2016, higher concentrations of 0.13 mg m$^{-3}$ of $NO_x$ and 1.7 mg m$^{-3}$ of CO were measured, a more than five-fold increase in average

daytime concentrations compared with the "clean" period. Corresponding increase and decreases in pollutant flux was not clearly observed however (fig. 3). Figure 5 shows the distribution of the $NO_x$ and CO fluxes and concentrations over the "clean" and "polluted" days. Despite the higher concentrations during the polluted period the measured flux is slightly lower. During the polluted period more deposition flux occurred; 13 % of $NO_x$ fluxes and 23 % of CO fluxes (associated with $u_*$ values over 0.175 m s$^{-1}$) were negative whereas no deposition fluxes were measured during the clean period.

Figure 6 shows two flux footprints; one averaged for the clean period and one for the polluted period. Satellite images and Open Street Map data (openstreetmap.org) shows that these footprints cover very similar land use areas; predominantly residential with busy roads so it is expected that the emissions would be similar for these areas, consistent with the variability in the measured flux. This indicates concentrations were more affected by meteorology, driving the accumulation at the city scale, or transport from regions outside Beijing than local emissions. There was not a significant contrast between mean mixing

layer heights between the two periods. A mean height of 464 m (including nighttime and daytime) with a daytime maxima of 811 m for the clean period was measured. For the polluted period the mean mixing layer height was 434 m with a daytime maxima of 822 m. Wind speeds were much lower during the polluted period with a mean wind speed of 1.9 m s$^{-1}$ compared to 6.0 m s$^{-1}$ for the clean period. These more stagnant conditions cause emissions to build up before being advected out of the city when wind speeds increase once again. Additionally, the higher wind speeds experienced during the "clean" period mean



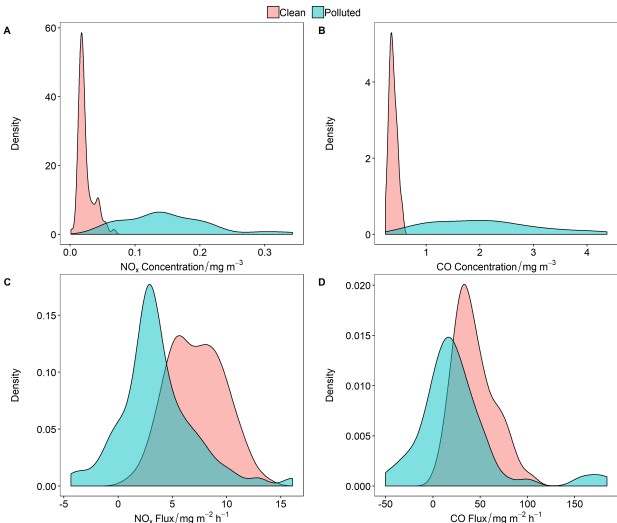

**Figure 5.** The density distribution of $NO_x$ and CO concentrations and fluxes during the "clean" period (21 November 2016 – 23 Novemeber 2016) and "polluted" period (24 November 2016 – 27 November 2016). Only fluxes for which $u_*$ values are over 0.175 m s$^{-1}$ and which meet the 60 % stationarity criteria are presented to allow a valid comparison.

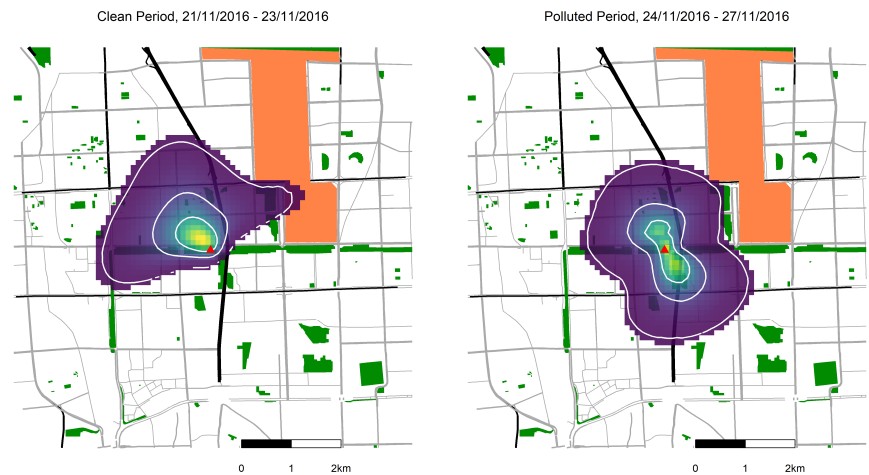

**Figure 6.** The left-hand plot plot shows the average footprint for the "clean" period, 21 November 2016 – 23 November 2016 and the right-hand plot the average footprint for the "polluted" period, 24 November 2016 – 27 November 2016. Map was built using data from © OpenStreetMap contributors 2019. Distributed under a Creative Commons BY-SA License.

the city is influenced by air masses from further away. To the north-west, the dominant wind direction for the "clean" period, there is less industrial activity compared to the south of the measurement site towards the centre of Beijing, so these air masses are also likely to be less polluted.





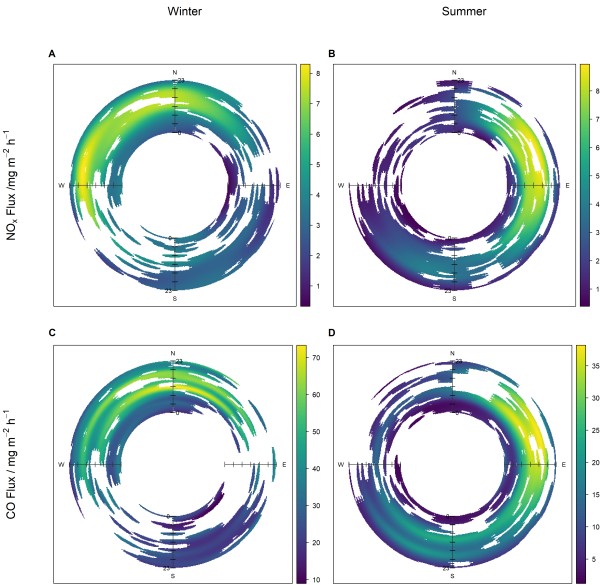

**Figure 7.** Polar annulus plots for $NO_x$ and CO fluxes for both campaigns, showing the relationship between mean diurnal flux and wind direction. 0 – 23 refers to hour of day and the colour scale shows flux in mg m$^{-2}$ h$^{-1}$.

### 3.3    Dependence on wind direction

Examining the relationship between diurnal flux and wind direction can give further information about emission sources. Figure 7 shows the average diurnal emission $NO_x$ and CO flux plotted as a function of wind direction for both the winter and summer campaigns. Hour of day is represented by the radial scale between the inner and outer rings and starts at 00:00 in the inner side

of the ring to 23:00 on the outer side of the ring.

For the winter campaign there appears to have been an enhancement in flux when there is a northerly wind direction. For $NO_x$ the emission was largest throughout the afternoon hours but for CO peaks are more distinct, with an enhancement at midday and again between 16:00 – 20:00. Within the flux footprint, to the north of the site are residential areas, Beitucheng West Road (a busy traffic route) as well as some university buildings and a hospital. Further north (approximately 1 km from

the tower) is the fourth ring road. The largest $NO_x$ fluxes were observed between 16:00 – 20:00 when the wind direction was westerly. Within this wind sector lies Beitaipingzhuang Road, about 0.65 km from the tower and Xueyuan Road which links the third and fourth ring roads slightly further to the west and just under 2 km away from the tower.

During the summer campaign, there was an enhancement in the daytime flux of CO and $NO_x$ with an easterly wind, indicating of an emission source to the east of the site; probably the Jingzang Highway just 0.35 km east of the site as shown in fig. 1. There

are also some daytime enhancements in $NO_x$ and CO emissions from the south of the measurement site towards central Beijing. For CO, enhancements to the south show two quite distinct enhancement periods in the morning and evening, suggesting these





**Table 2.** Summary table for VOC concentrations and fluxes measured by PTR-MS. Data presented is for fluxes which are within 60 % stationarity criteria for all $u_*$ values.

| | Winter | | | | Summer | | | |
|---|---|---|---|---|---|---|---|---|
| **Conc (mg m$^{-3}$)** | **Benzene** | **Toluene** | **C2-Benzene** | **C3-Benzene** | **Benzene** | **Toluene** | **C2-Benzene** | **C3-Benzene** |
| Mean | 0.00703 | 0.00808 | 0.00975 | 0.00258 | 0.00186 | 0.00257 | 0.00390 | 0.00127 |
| Median | 0.00532 | 0.00620 | 0.00688 | 0.00198 | 0.00173 | 0.00227 | 0.00336 | 0.00113 |
| Percentiles | | | | | | | | |
| 5th | 0.000910 | 0.00100 | 0.00121 | 0.000431 | 0.000860 | 0.00113 | 0.00166 | 0.000572 |
| 95th | 0.0190 | 0.0225 | 0.0299 | 0.00658 | 0.00346 | 0.00502 | 0.00794 | 0.00245 |
| Standard Deviation | 0.00618 | 0.00707 | 0.00969 | 0.00198 | 0.000864 | 0.00227 | 0.00237 | 0.000677 |
| **Flux (mg m$^{-2}$ h$^{-1}$)** | | | | | | | | |
| Mean | 0.0121 | 0.0640 | 0.0730 | 0.00357 | 0.101 | 0.307 | 0.236 | 0.148 |
| Median | 0.0197 | 0.00861 | 0.00656 | 0.00354 | 0.0858 | 0.197 | 0.131 | 0.0995 |
| Percentiles | | | | | | | | |
| 5th | -0.296 | -0.209 | -0.270 | -0.0662 | 0.0180 | 0.137 | -0.0166 | 0.0207 |
| 95th | 0.195 | 0.427 | 0.692 | 0.0737 | 0.237 | 0.872 | 0.941 | 0.479 |
| Standard Deviation | 0.184 | 0.249 | 0.286 | 0.0455 | 0.0736 | 0.363 | 0.374 | 0.151 |

emissions are from vehicles. It is unlikely that there were any significant seasonal change in traffic density on roads surrounding the measurement site. The change from a westerly influence observed during the winter to an easterly influence in summer is due to changes in the dominant wind direction. Beijing's wind patterns are quite different during the winter and the summer. There is a noticeable absence of easterly winds in the winter and the mean flux footprints (fig. 2) highlights the difference in

regions contributing to the observed flux.

### 3.4 Comparison with VOC Flux

Aromatic hydrocarbons, including benzene, $C_2$-benzene, $C_3$-benzene and toluene are components of gasoline fuel and are typically emitted from combustion and evaporation of fuels and solvents in urban environments (Caplain et al., 2006; Langford et al., 2009). Table 2 summarises the fluxes and concentrations measured during the APHH-Beijing measurement campaigns.

Concentrations were significantly greater during the winter season while emissions were higher during the summer for all four hydrocarbon species. During the summer, toluene fluxes were the largest (0.31 mg m$^{-2}$ h$^{-1}$), followed by $C_2$-benzene (0.24 mg m$^{-2}$ h$^{-1}$) then $C_3$-benzene (0.15 mg m$^{-2}$ h$^{-1}$) and the flux of benzene being the smallest (0.10 mg m$^{-2}$ h$^{-1}$). Diurnal variation of VOCs for winter and summer is shown in fig. 8. The uncertainty in the diurnal averages makes drawing conclusions about variations between daytime and nighttime difficult, although there are some indications that summertime trends at least may

mimic those of CO and NO$_x$. During summer, fluxes of all four species followed a very similar diurnal trend with generally higher emissions in the daytime compared to the nighttime. A smaller early morning peak was observed at 08:00, slightly later





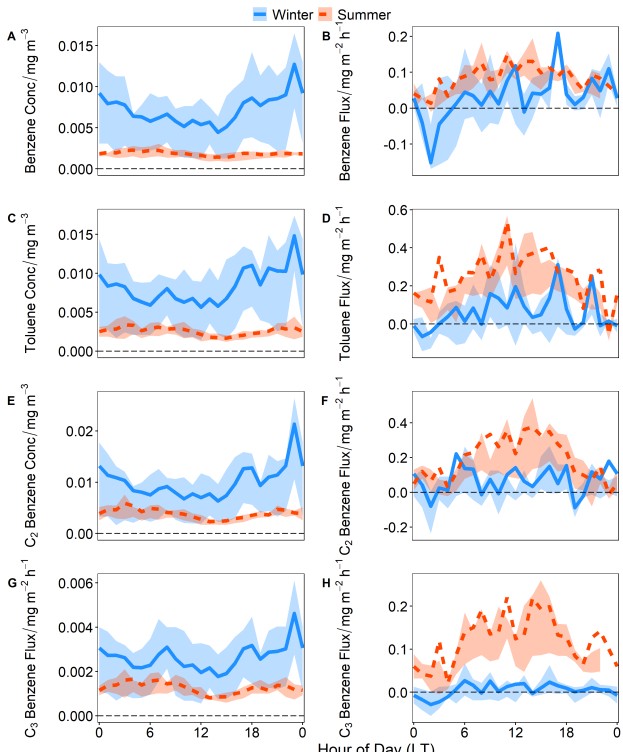

**Figure 8.** Seasonal variation in diurnal profile for A) benzene concentration, B) benzene flux, C) toluene concentration, D) toluene flux, E) $C_2$-benzene concentration, F) $C_2$-benzene flux, G) $C_3$-benzene concentration and H) $C_3$-benzene flux measured by PTR-MS. Blue, solid lines are for measurements taken during the winter campaign and the orange, dashed lines are measurements taken during the summer campaign. The shaded areas represent the $25^{th}$ and $75^{th}$ percentiles to give an idea of the spread of the data. Only stationary data has been used but no $u_*$ filtering has been applied when doing the diurnal averaging.

than the morning peak in $NO_x$ emissions which reached their maximum at 07:00. Emissions increased again in the afternoon for toluene, $C_2$-benzene and $C_3$-benzene peaking between 15:00 – 16:00. Benzene remained fairly constant throughout the afternoon with a mean emission of 0.076 mg m$^{-2}$ h$^{-1}$. The peak during the night corresponds to a one-off event measured on the 29/05/2017 at 03:30. During the winter, $C_2$-benzene fluxes were the largest (0.073 mg m$^{-2}$ h$^{-1}$), followed by the toluene flux (0.064 mg m$^{-2}$ h$^{-1}$), then benzene (0.012 mg m$^{-2}$ h$^{-1}$) with the flux of $C_3$-benzene flux (0.0036 mg m$^{-2}$ h$^{-1}$) being the smallest. There was no clear difference between nighttime and daytime emissions for all species.

Table 3 summarises previous urban flux measurements to allow the VOC fluxes from Beijing to be put into context. The benzene fluxes measured in Beijing during summer fall into the range of those reported in table 3 and are most similar to values reported for large cities in the UK. Toluene fluxes and $C_2$-benzene fluxes measured during the summer in Beijing are also similar to the mean fluxes measured in UK cities where vehicular emissions were found to dominate aromatic fluxes. However, it should be noted that previous measurements over London were all made with a quadrupole PTR-MS instrument,





**Table 3.** Summary of mean VOC fluxes measured in various urban or semi-urban locations.

| VOC Species | Mean Flux /mg m$^{-2}$ h$^{-1}$ | Location and Year | Reference |
|---|---|---|---|
| **Benzene** | 0.0121 | Beijing (November – December 2016) | *This study* |
| | 0.101 | Beijing (May – June 2017) | *This study* |
| | 4.7* | Mexico City, March – April 2006 | Karl et al. (2009) |
| | 0.396 | Mexico City, March 2006 | Velasco et al. (2009) |
| | 0.12 | Manchester, June 2006 | Langford et al. (2009) |
| | 0.15 | London, October 2006 | Langford et al. (2010) |
| | 0.09 | London, August – December 2012 | Valach et al. (2015) |
| | 0.02 | Helsinki (urban background), January 2013 – September 2014 | Rantala et al. (2016) |
| | 0.07* | Central London, July 2013 | Vaughan et al. (2017) |
| **Toluene** | 0.0.0640 | Beijing (November – December 2016) | *This study* |
| | 0.307 | Beijing (May – June 2017) | *This study* |
| | 0.83 | Mexico City, April 2003 | Velasco et al. (2005) |
| | 14.1* | Mexico City, March – April 2006 | Karl et al. (2009) |
| | 3.1 | Mexico City, March 2006 | Velasco et al. (2009) |
| | 0.28 | Manchester, June 2006 | Langford et al. (2009) |
| | 0.41 | London, August – December 2012 | Valach et al. (2015) |
| | 0.051 | Helsinki (urban background), January 2013 – September 2014 | Rantala et al. (2016) |
| | 0.24* | Central London, July 2013 | Vaughan et al. (2017) |
| **C2-Benzene** | 0.0730 | Beijing (November – December 2016) | *This study* |
| | 0.236 | Beijing (May – June 2017) | *This study* |
| | 0.468 | Mexico City, April 2003 | Velasco et al. (2005) |
| | 1.3 | Mexico City, March 2006 | Velasco et al. (2009) |
| | 0.32 | Manchester, June 2006 | Langford et al. (2009) |
| | 0.28 | London, October 2006 | Langford et al. (2010) |
| | 0.059 | Helsinki (urban background), January 2013 – September 2014 | Rantala et al. (2016) |
| | 0.32* | Central London, July 2013 | Vaughan et al. (2017) |

*Measured from an aircraft.*





which, unlike the time-of-flight instrument used in the present study, is restricted to a unit mass resolution. This means that the benzene, toluene and $C_2$-benzene signals may have included contributions from other compounds and therefore is likely to be an overestimate of the true $C_2$-benzene emission. The VOC flux values reported for Mexico City are much greater than those measured in Beijing for both seasons. Karl et al. (2009) reported an influence from evaporative emissions from the northern

industrial district but there is no industrial emission source within the flux footprint for the Beijing measurements. This, along with a more modern vehicle fleet subject to stricter emissions regulation in Beijing may explain the larger discrepancy between these two cities. Fluxes measured in Helsinki are lower than emissions measured in the summer but comparable to those measured during the winter.

Emissions of all four species were higher in summer than in winter, although the statistical significance of all but $C_3$-
benzene is somewhat uncertain. This is the opposite to the trend observed for $NO_x$ and CO emissions, most likely driven by higher evaporation rates due to higher temperatures. The mean daytime temperature was 281 K during the winter campaign and 305 K during the summer campaign. Comparing VOC fluxes to $NO_x$ and CO fluxes in the summer, emissions of $C_2$-benzene and $C_3$-benzene started to increase in the early hours of the morning (05:00) as observed for $NO_x$ and CO, indicating a rapid release of emissions from a new source, such as traffic emissions. This is true for toluene and benzene, although the relative
difference between night and day emissions is less pronounced. All four hydrocarbon species peak showed an enhancement during the mid-morning during the summer, like CO, probably due to an additional contribution from residential and cooking emissions from local restaurants. There is possibly a winter contribution from evening rush hour traffic to the benzene and toluene emissions that mirrors that observed in the CO and $NO_x$ emissions at 17:00, though this is not clearly observed during the summer campaign. During the summer, the emission for all four species are elevated in the afternoon (between 13:00 –
18:00) when temperatures are highest. These emissions are likely enhanced by evaporation which masks the evening rush hour traffic contribution. There is a strong correlation between total VOC flux and heat flux in summer ($r = 0.83$) compared to the winter ($r = 0.35$).

The ratio of benzene to toluene (B/T) is often used to gauge the photochemical age of an air mass as the two species have different atmospheric lifetimes due to their different reactivities with the OH radical. Heeb et al. (2000) reported B/T
concentration ratios between 0.41 – 0.83 for primary exhaust emissions. As an air mass ages, the ratio increases as toluene is more reactive than benzene. For Beijing, the median B/T concentration ratio in the winter was 0.89 and 0.73 for the summer at the upper end of the expected range for primary exhaust emissions. Barletta et al. (2005) reported a roadside B/T value of 0.6 for Beijing, however vehicle fleet and fuel types are rapidly changing in response to legislation so this measurement may not be representative for the measurement period of this work. The difference in B/T concentration ratios does however indicate that
VOC sources are changing between winter and summer as the change in ratio cannot be explained by temperature differences. Langford et al. (2009) points out that temperature can impact B/T ratios as reactivity rates increases with warmer temperatures but if temperature was driving the seasonal difference it would be expected that B/T ratio measured during summer is higher than during winter. It is likely that additional sources such as domestic burning and cooking are present in winter; Barletta et al. (2005) reports that higher B/T concentration ratios are associated with natural gas and biomass combustion, which supports
the hypothesis that these type of emission sources could be leading to an increased B/T concentration ratio in the winter. The





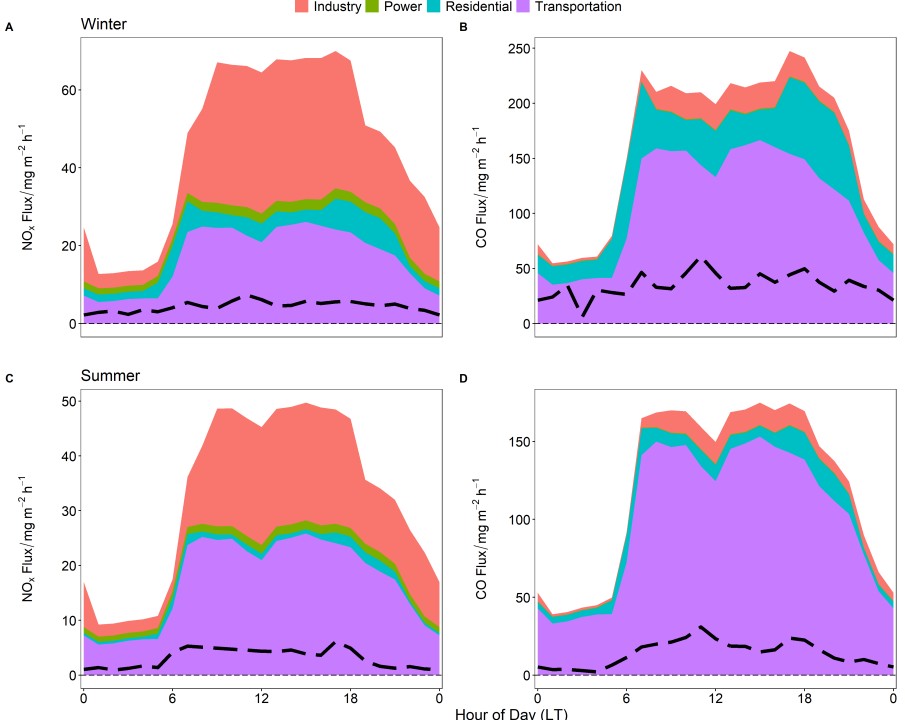

**Figure 9.** Comparison of measured diurnal trend and diurnal trend predicted by the MEIC inventory for $NO_x$ and CO. Panels A and B show $NO_x$ and CO emissions for the winter campaign and panels C and D $NO_x$ and CO emissions from the summer campaign. Four sections of the inventory are presented alongside the measured emission as a black dashed line.

B/T flux ratio should reflect better the ratio of the two pollutants in emissions sources as flux ratios are confined to the area of the flux footprint rather than being influenced by a wider area (Karl et al., 2009). The median B/T flux ratios were 0.72 for the winter campaign and 0.31 for the summer campaign, further suggesting a change in emission source between the two seasons. The B/T flux ratio for summer is lower than expected for primary exhaust emissions though similar values have been reported for vehicles without catalytic converters (Heeb et al., 2000) and given higher temperatures, emissions from other sources such as solvent evaporation, may affect this ratio.

### 3.5 Comparison with an Emissions Inventory

The measured $NO_x$ and CO fluxes were compared to the high-resolution (3 km × 3 km) MEIC v1.3 inventory and the mean diurnal profiles are presented in fig. 9. The comparison of VOC emissions with an emissions inventory is beyond the scope of this work. For both winter and summer, the inventory grossly overestimates emissions of $NO_x$ and CO throughout the day. Total $NO_x$ emissions are overestimated by a factor of 3.8 – 17 (mean overestimation of 9.9 throughout the day) in the winter and 4.2 – 25 (mean = 11) in the summer. For CO, winter emissions were between 1.6 – 9.7 (mean = 4.8) times larger in the inventory than those measured and summer emissions between 5.2 – 21 (mean = 10) times larger.





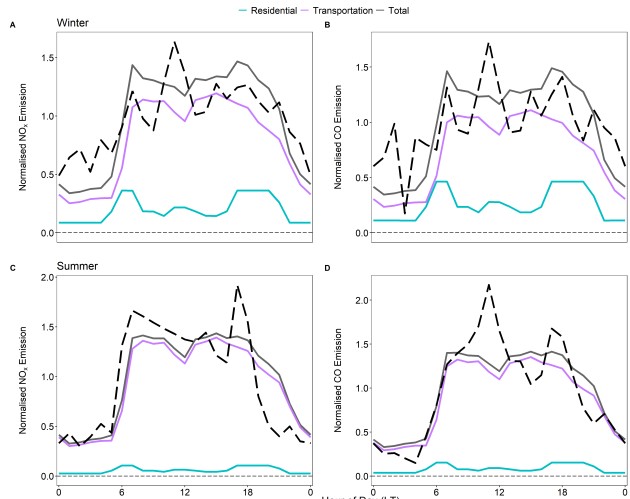

**Figure 10.** Comparison of normalised diurnal variation in $NO_x$ and CO emission predicted by the MEIC inventory and measured flux. The normalised emission has been calculated for the inventory by dividing hourly sector emission by the mean daily sector emission. Given there are no clear industrial or power generation emission sources within the flux footprint, only the residential (blue solid line) and transportation (purple solid line) sectors have been presented. The normalised diurnal variation for these sectors have been multiplied by the normalised diurnal variation for the sum of the residential and transportation sectors (grey solid line) to reflect the relative contributions of the two sectors. Normalised diurnal variation in the measured emission has been calculated by dividing mean hourly emission by mean daily emission (black dashed line). Panels A and B show data for $NO_x$ and CO from the winter campaign and panels C and D for the summer.

Examination of the flux footprint suggests that the majority of measured emissions are coming from transportation and residential sources. The inventory supports this for CO, suggesting that transportation is the largest contributing sector, followed by the residential sector. However, for $NO_x$ the inventory also suggests that there is a large industrial source, contributing up to 60 % to the total emissions for the winter campaign and 52 % for the summer campaign. No obvious industrial sources could

be identified within the flux footprint for this study. Zheng et al. (2017) observed a decoupling between real world emissions and the spatial proxies used to develop inventories as polluting industries are moved out of urban centres. The methods used to allocate emissions in the MEIC inventory appear to result in emissions being overestimated in the urban area of Beijing; spatial proxies such as population density and Gross Domestic Product (GDP) are used to scale down national emissions statistics. This method tends to overestimate emissions in urban centres and underestimate emissions in rural areas and so even

if this comparison only considered the residential and transport sectors, $NO_x$ and CO emissions would still be significantly overestimated by the inventory. Just considering these two sectors, $NO_x$ emissions are overestimated by a factor 2.4 – 7.3 (mean = 4.7) for winter and 3.4 – 15 (mean = 6.5) for summer. CO emissions are overestimated by 1.5 – 9.2 (mean = 4.4) in winter and 4.7 – 20 (mean = 9.2) in summer. The sensitivity of the inventory to location was tested by shifting the inventory grid 3 km in each direction (north, east, south and west) and it was found that this had little impact on the comparison with the

inventory still overestimating emissions for all directions.





The inventory did capture some of the general diurnal variation in emissions. Figure 10 shows the normalised diurnal variation for the transportation and residential sectors, calculated relative to one another assuming these sectors are the only emission sectors. The diurnal variation in measured emissions normalised by the daily average is overlaid. For NO$_x$ and CO emissions in the winter (fig. 10, panels A and B) the sum of the residential and transportation emissions captures the time of the morning rush hour peak at 07:00 and the evening enhancement in emissions in the mid-afternoon to evening due to a combination of increased residential and traffic activity. Residential emissions are predicted in the inventory to increase between 11:00 – 12:00 which was observed in the measurement though to a greater extent than the inventory suggests.

Looking at the summer data for NO$_x$ (fig. 10, panel C), the increase in emissions after 05:00 when traffic density increased was well captured for NO$_x$ though the measurement suggested a slightly quicker increase in emission than suggested by the inventory. The minimum daytime emission predicted by the inventory occurred at 12:00 where there was a dip in emissions from transportation though this was not reflected in the measurements where there was an almost constant decrease in NO$_x$ emissions until the evening rush hour peak at 17:00. The evening rush hour peak for NO$_x$ in summer was much more distinct than the inventory suggests and NO$_x$ emissions decrease rapidly after 18:00, earlier than the inventory predicts. NO$_x$ emissions from residential sources in the inventory showed a small enhancement around midday which was not clearly observed in the measurement, suggesting traffic related emissions dominate here.

For CO in summer (fig. 10 panel D), the increase in measured emissions after 05:00 matches the rate of increase predicted the inventory very well. After 07:00 the inventory predicted CO emissions would not increase further which was not reflected in the measured emissions. Emissions from the residential sector are predicted to increase between 11:00 – 12:00 reflecting emissions from cooking activities. Within the flux footprint there were several restaurants including barbecue restaurants where food was cooked over open coals. This is not likely to be included in the inventory but would explain the increase in measured CO emissions before lunchtime as food is prepared. As with NO$_x$ emissions, the evening rush hour peak in measured CO emissions was narrower and began to decrease earlier than suggested by the inventory.

The inventory suggests that total emissions are higher in winter than in summer; NO$_x$ emissions from the residential and transportation sectors are between 1.1 – 1.4 times larger and for CO emissions from these two sectors are between 1.2 – 1.6 times larger. The inventory performs relatively well capturing the measured seasonal difference in NO$_x$ (emissions were on average 1.2 times greater in winter compared to summer) but not for CO (emissions were on average 2.3 times higher in winter than summer). Inventory NO$_x$ emissions for transportation are almost identical for the two seasons whilst residential emissions vary by a factor of four. The inventory attributes seasonal differences in CO emissions chiefly to the residential sector, estimating approximately a 4 fold increase in residential emissions in winter compared to summer. CO emissions from the transportation sector are between 1.05 – 1.09 times higher in winter than summer suggesting any effects of vehicle cold starts and reduced engine combustion efficiencies are not represented in the inventory; something that was considered the main source of seasonal variation in CO fluxes in London (Helfter et al., 2016).

Aside from the error introduced through the use of the spatial proxies for scaling, discrepancies between inventory and measurements are likely to be due to the comparison of observations (2016/2017) with an older inventory (2013). China's NO$_x$ emissions have rapidly changed in the past three decades. Liu et al. (2016) report that between 2005 to 2011 NO$_2$ emissions





increased by 53 % for the whole of China, attributed for the most part to increasing fuel consumption with coal the dominant fuel type. An estimated three-quarters of all electricity in China was generated by coal in 2016 (International Energy Agency, 2016). After 2012 however, a combination of the installation of power plant de-nitration devices and vehicle emissions controls has led to a 32 % decrease in $NO_2$ emissions (Liu et al., 2016; Krotkov et al., 2016; Miyazaki et al., 2017). In Beijing, $NO_x$

emissions are decreasing thanks to numerous air pollution control measures implemented since 2000; polluting industries and power plants have been relocated outside of the city, stricter emissions standards for industrial and domestic boilers have been introduced and the fuel type shifted from coal to gas (Wang et al., 2010). Since the introduction of the "Clean Air Action Plan" in Beijing in 2013, 900,000 households in Beijing have converted from using coal to cleaner technologies like gas or electricity. The burning of biomass, such as wood and crops, was completely forbidden by the end of 2016 (Cheng et al., 2019).

The impact of the emissions controls has been predicted to reduce emissions of $NO_x$ and VOCs by 43 % and 42 % respectively between 2013 – 2017 in Beijing (Cheng et al., 2019). Most significant for $NO_x$ emissions however is the stringent vehicle control measures introduced within the last decade, accounting for 47 % of the total reduction in emissions for the city.

CO emissions have also been declining in Beijing over the past two decades by an average rate of 1.14 % year$^{-1}$ (Wang et al., 2018). Zheng et al. (2018) highlight a reduction in inefficient domestic stoves and improvements in emissions standards

for vehicles as being dominant forces for the observed reduction in CO emissions. For VOCs, vehicle emission controls were another significant contributor to the reduction with 16.1 % of the reduction attributed to new controls. Improvements in management of solvent use (e.g. use of high-solid and waterborne paints instead of solvent based ones) dominated the reduction in VOC emissions contributing 49.3 % to the total reduction in Beijing. Emissions between 2013 – 2016 were predicted to reduce by 30 % for $NO_x$ and 35 % for VOCs (Cheng et al., 2019; Biggart et al., 2019).

Taking these reductions into account and lowering the inventory emissions by 30 % in the winter and 43 % for the summer $NO_x$ is still overestimated by a factor of 1.6 – 7.2 in the winter and 1.6 – 9.0 times in the summer. No emissions reductions were presented for CO alone but given that its expected sources are similar to $NO_x$ the same reduction factor was applied to the CO inventory emissions. With this reduction taken into account CO emissions predicted by the inventory were at times lower than those measured during the night, with the inventory CO emissions being 0.84 – 5.0 times the measured CO emission. For

summer, the inventory overestimated CO emissions by a factor of 2.2 – 9.0 times. The closest agreements between inventory and measurements were during the nighttime in all cases. Applying these reductions does obviously improve the inventory comparison however given the large overestimations it is likely that the uncertainty caused by use of spatial proxies when developing the inventory is the main one.

## 4 Summary

$NO_x$, CO and aromatic VOC emissions have been quantified for the first time during two contrasting seasons for an area of central Beijing. The magnitude of $NO_x$ emissions were found to be similar during the winter and summer periods whilst the fluxes of CO showed a greater seasonal dependence with winter emissions being over 2 times greater than CO emissions measured during summer. The dominant source for $NO_x$ and CO emissions is traffic with influence from residential emissions.





The diurnal variation in aromatic VOCs fluxes also suggested traffic and residential sources, though evaporative effects due to higher summer temperatures meant emissions were greater in the summer than the winter. The $NO_x$ and CO fluxes presented in this work provide good evidence that proxy-based inventories can overestimate emissions for urban centres as suggested by Zheng et al. (2017). When developing inventories at an urban scale future work should look carefully at up-to-date proxies or at deriving emissions by bottom-up approaches in order to correctly predict the magnitude of emissions. When comparing diurnal variation in inventory and measurement the inventory performed relatively well capturing morning and evening rush hour peaks. The inventory also attributed traffic and residential emissions as the major source of $NO_x$ and CO emissions which was supported by analysis in this work. This set of pollutant flux measurements can provide a useful basis for developing these high resolution urban inventories which are at an appropriate scale to assess public health impacts of pollution.





**Appendix A**

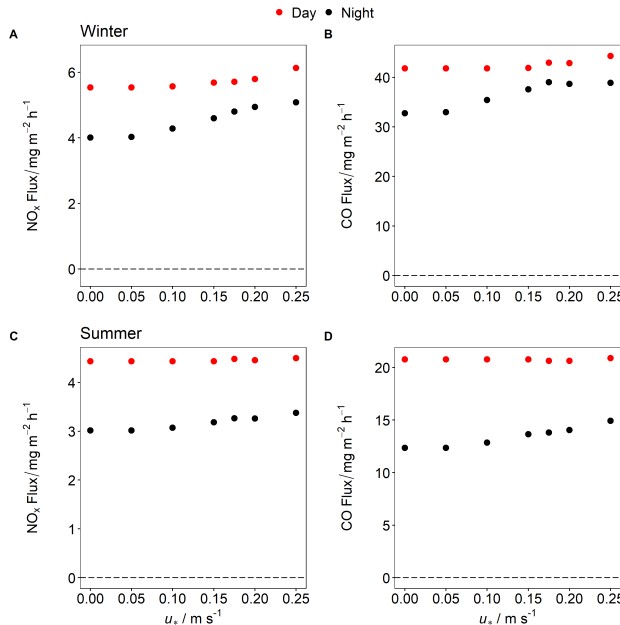

**Figure A1.** Mean $NO_x$ and CO fluxes as a function of different $u_*$ thresholds.





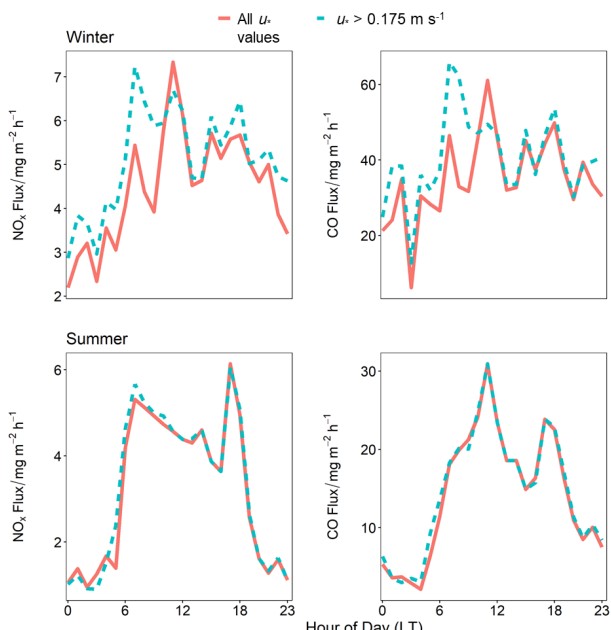

**Figure A2.** Comparison between diurnal variation in $NO_x$ and CO fluxes for all $u_*$ values and for $u_*$ values over 0.175.

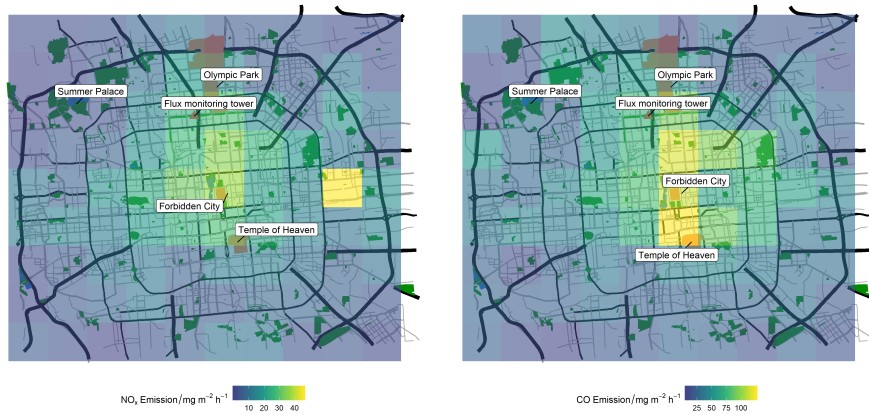

**Figure A3.** MEIC inventory emission grids for $NO_x$ and CO for November. Map was built using data from © OpenStreetMap contributors 2019. Distributed under a Creative Commons BY-SA License.





*Author contributions.* FAS made $NO_x$ and CO concentration measurements, calculated their flux and analysed the data presented. FAS prepared the manuscript with contributions from co-authors. EN and BL measured wind vector data used in this study, set up tower instrumentation, provided extensive advice on flux calculations and provided detailed comments on the manuscript. OW assisted with interpretation of the inventory emissions data and gave ideas for analysis. WSD provided support calculating $NO_x$ and CO fluxes and reviewed the manuscript.

5  WJFA and MS made VOC concentration measurements during the field campaigns and assissted with interpretation of the VOC fluxes. MH processed the raw emissions data into gridded format for comparison with the measured fluxes. SBG and SK provided boundary layer height data. ND and SM provided initial support on using eddy4R software for data processing and reviewed the manuscript. QZ and RW provided high resolution emissions data. XW and YZ prepared the PTR-MS instrument and calibration system. PF maintained the tower and site necessary for this work. CNH, JFH and JL reviewed the manuscript.

10  *Competing interests.* The authors declare that they have no conflict of interest.

*Acknowledgements.* This work was supported by the UK Natural Environment Research Council and the Newton fund through the AIR-POLL project of the Air Pollution and Human Health in a Chinese Megacity (APHH-Beijing) programme (grant references NE/N006917/1, NE/N006992/1 and NE/N006976/1). The authors would like to thank Rachel Dunmore and Neil Mullinger for their hard work during the field campaigns. Thanks also go to Shona Wilde and Stuart Grange for their assistance creating plots for this work.



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
