# Peer review of "Measurements of traffic dominated pollutant emissions in a Chinese megacity."

_Atmospheric Chemistry and Physics, 2019_

## Referee Comment (RC1) · Anonymous Referee #1 · 13 Feb 2020

General comment: The work by Squires et al. presents traffic related pollutant emissions in an Asian megacity. It presents novel data for a globally important region struggling with air quality. The manuscript is appropriate for publication in ACP after addressing some of the more specific comments outlined below.

Specific comments: Section 2.4: It is not entirely clear how the data treatment is done. Were the data time-shifted before submitting these to the eddy flux routine or did the eddy flux routine take account this shift and perform a covariance analysis? A better approach to deal with lag-time estimation when individual 30 min covariance functions are below the LOD (to estimate a lag time) is to average quality filtered covariance functions. This allows to obtain a study (or weekly) average lag time for each individual species. This is a more accurate approach if compounds exhibit different absorption

[Figure]

and desorption properties along an inlet line.

Section 2.5: Line 6, 16: The authors should outline the changes incorporated in eddy4R that were specifically implemented and necessary for the present study (e.g. by showing a work flow diagram). Flux footprint model. The flux footprint model is based on Kljun et al., 2004 and was modified according to Metzger et al. 2012. Kljun et al. 2012 updated their original footprint model. It is not clear what the exact differences between the cross-integrated footprints between Kljun et al. 2012 and Metzger et al. 2012 are, assuming they are not the same. A clarification of this issue would be warranted.

Section 2.7: Page 9: "For this evaluation, an optimized version of the MEIC v1.3 inventory for 2013 was used that was derived by iňĄtting the NAQPMS model with observed pollutant concentrations during the campaign periods." What was done explicitly to optimize the MEIC inventory here. The authors cite a reference, but it would help the reader to understand the approach if more information on the procedure was given here.

Section 3.1: In this manuscript fluxes are generally reported as mg/m2/h. How was the NOx flux derived in units of mg/m2/h? The NOx channel would strictly only allow to report fluxes on a molar basis. Partitioning NOx and NO fluxes from the two channels could introduce additional uncertainty. Tower setup: at the height of the tower one would expect that the measurements are decoupled from the surface and represent the residual layer during night. How is this taken into account into the interpretation of night time data? Could this influence the storage flux calculation for night time?

Table 1: Additional measurements where BTEX fluxes were reported directly should be included in Table 1. For example, Karl et al. 2018 (10.1073/pnas.1714715115) report 24h average benzene (toluene) fluxes of 20 (82) ug/m2/h for Innsbruck. Park et al., 2010 (10.1016/j.atmosenv.2010.04.016) present BTEX fluxes for Houston with maximum daytime fluxes in the range of 0.2-0.3 mg/m2/h and 0.5 -0.7 mg/m2/h.

Editorial comments: Page 1: line 1: fluxes? Page 2: line 12 cc PM can also be emitted by primary sources, depending on size, primary or secondary production is more relevant. As Chinese efforts to reduce primary PM are regarded successful, a reference could be given to what extent secondary aerosols are nowadays dominating in a city like Beijing Page 2: line 15 is a repetition of what was said a couple of lines earlier – it could be rewritten more concisely Page 2: line 18: "At" high concentration Page 5: line 17: . . . was also. . . . ! Page 6: line 7 cc: this is not a complete sentence. Page 7: line 17: Fig. A1, Fig. A2. . . . Page 8: line 10: . . ..tower tower. . .. Page 8: line 30:. . .. to THE measured flux ? General editorial comments. Naming and formatting of figures, tables and references should follow the ACP editorial guidelines and should be copy edited. For example many references are cited as" by (Famulari et al, 2010;). . .." Probably due to the citation software used. ACP formatting guidelines however suggest to cite as following: by Famulari et al. (2010). Similarly, references to figures do not follow ACP formatting guidelines and should be corrected.

---

## Referee Comment (RC2) · Anonymous Referee #2 · 26 Apr 2020

Squires et al. describe seasonal differences in flux measurements of CO, NOx, and select VOCs in Beijing during winter and summer, 2016-2017. The authors show that seasonal differences in combustion markers can be largely attributed to changing sources between the two seasons (residential vs. transportation), while changes in the fluxes of aromatic VOCs are strongly influenced by evaporative emissions during summer. The authors compare the flux measurements to inventory estimates from the MEIC, and discuss the significant discrepancies between the measurements and the inventory. The authors also use the inventory to infer how seasonal differences in emission sectors may have contributed to the observed seasonality in CO and NOx fluxes.

Overall, the manuscript is very well written and organized, and the authors tell a compelling story as to what contributes to the CO, NOx, and VOC sources observed in Beijing. It is also an important contribution to constraining emission inventories in China, which evolve quickly as sources continue to face regulations. The manuscript should be published in ACP, provided that the authors address a few comments pertaining to the comparison of the measurements and inventory.

**Main Comments**

1. Section 3.4. I appreciate the authors' careful discussion and source attribution of the VOC fluxes, since this can be more complicated than for CO and NOx due to the variety of processes that could contribute to enhanced VOC fluxes (emissions, temperature, etc.). I am impressed by the general higher flux of aromatics in the summer and good correlation with heat flux. This seems to suggest that evaporative emissions are a driving factor in the behavior of the aromatics (as concluded by the authors), and could be the primary explanation in the seasonal differences in B/T ratios. The authors note that a detailed comparison of the VOC emissions with the inventory is beyond the scope of the work (which is reasonable), but is it possible to do a comparison between the aromatic / benzene ratios measured in this work with VOC profiles represented in the MEIC or other inventories? Li et al. (2017) tends to show that residential uses of aromatics result in higher emissions than the transportation sector. Perhaps a comparison of the B/T ratio between sectors would provide additional evidence as to what is contributing to these seasonal differences.

2. Section 3.5 and Figure 9. In this section, the authors discuss the measurements in context with the MEIC. The discussion of this section largely hinges on Figure 9, which suggests that the inventory largely overestimates NOx and CO emissions. In my opinion, this figure is misleading, since this is comparing an inventory from 2013 to measurements conducted in 2017. The authors note later in the section that the NOx and CO emissions in China change drastically on a multi-year basis. The authors discuss how these changes would have likely affected the MEIC estimates from 2013, but still conclude that the inventory is overestimated.

   I think that it would be most fair to structure this section to first show how the

inventory would have changed from 2013 until 2017, then adjust the inventory estimates appropriately to account for these CO and NOx reductions, and finally compare these profiles to the measurements conducted in 2017 and discuss the discrepancies. Otherwise, a reader who skims the figures might conclude that the inventory is entirely out to lunch, which isn't truly the case.

**Other Comments**

3. Page 2, Lines 25 - 30. The authors note that total emissions from inventories can differ from measured fluxes but I think it is also important to note that the distribution of VOCs can be different due to unattributed sources. Karl et al (2018) shows that there is a large, unidentified source of oxygenated VOCs (OVOCs) in European cities that isn't properly represented in emission inventories. The same conclusions were drawn by McDonald et al. (2018), who showed that these sources likely result from solvent emissions due to the use of consumer and industrial products. As China places stricter restrictions on emissions, it's likely that the relative importance of different emissions sectors will also change. Could the authors provide some context here (as they do in other places in the manuscript), and perhaps provide a brief overview of the current VOC, NOx, and CO distribution of China's emission inventory?

4. Section 2.3. How (and how frequently) were background PTR-MS measurements performed? The authors might consider referring to the instrument as a PTR-ToF-MS, since this is not a quadrupole instrument (the original PTR-MS). This distinction is important because the high time resolution of a PTR-ToF-MS is needed for flux calculations.

5. Page 5, Line 30. "Despiked" is an odd term. Would "smoothed" be a better option?

6. Page 6, Line1: "allowing the software to calculate" is very vague. Was there an algorithm applied to the data to determine instrument lag in real time?

7. Page 6, Line 23. What do the authors mean by "non-stationary periods"? Does this mean that the flux changing too rapidly over the course of the averaging period?  This statement seems repetitive with the previous sentence.

8. Figure 3. It is difficult to see the grey colored traces highlighting non-stationary periods, and it's also confusing to have the contrasting periods also highlighted with grey. I would recommend adding markers for the points that were non-stationary, changing the color t clearly indicate these points, or simply remove these from the figure.

9. Page 13, Line 5 and Line 23. The authors mention that the major NOx sources are likely the same between the two seasons, which is likely true for vehicle traffic. However, wintertime fluxes appear to be higher on average (~20%), and the wintertime diurnal

patterns shows little resemblance to the summertime pattern (in line with the differences seen for CO). Furthermore, the nighttime flux of NOx seems to be substantially higher in the winter. To my eye, this suggests that there are additional important NOx sources, which are probably the same as those implicated for CO (i.e., residential heating). This seems to be supported by Fig 10, which shows that the Residential/Transportation ratio for NOx is substantially larger in the winter than compared to summer.

10. Page 13, Line 33. What is meant by "distant heating"? Is this heating using electricity?

11. Page 13, Lines 34 - Page 14, Line 6. The purpose of this paragraph isn't entirely clear. Are the authors invoking HDV emissions to explain the higher CO and NOx emissions during winter, or is this to explain the nighttime emissions during both seasons? Would one expect HDV's to be more active during winter than during summer? It does appear that the summertime morning peak in NOx, which is earlier than the CO peak, could be associated with the higher truck traffic before 06:00.

12. Figure 7: This figure is very nice. It is a little difficult to read the labels and to see the time axis on top of the darker colors of the fluxes. Could these be presented in larger fonts, and in bold?

13. Page 20, Line 30. When the authors mean that the change in B/T cannot be explained by temperature differences, I assume they mean atmospheric oxidation and not evaporation processes. This is clarified in the following sentence, but it would be good to be precise here since there is a lot of discussion about what could be affecting the B/T ratio (evaporation, chemistry, etc.) and mixing terms is a bit confusing.

**References**

Li et al. (2017). MIX: a mosaic Asian anthropogenic emission inventory under the international collaboration framework of the MICS-Asia and HTAP. Atmos. Chem. Phys., 17, 935-963.

McDonald et al. (2018). Volatile chemical products emerging as largest petrochemical source of urban organic emissions. Science, (359) 6377, 760-764.

---

## Author Comment (AC1) · 27 May 2020

The comment was uploaded in the form of a supplement:
https://www.atmos-chem-phys-discuss.net/acp-2019-1105/acp-2019-1105-AC1-
supplement.pdf
* * *

---

## Author Response (AR1)

This document includes authors' responses to both reviewer 1 and reviewer 2. Reviewer's comments are in bold text. Authors' responses are plain text with any text taken from the manuscript in italics. New additions to the manuscript in response to reviewer comments are shown in red italics.

**Response to RC1: Anonymous Referee #1**

General comment: The work by Squires et al. presents traffic related pollutant emissions in an Asian megacity. It presents novel data for a globally important region struggling with air quality. The manuscript is appropriate for publication in ACP after addressing some of the more specific comments outlined below.

The authors thank the reviewer for the supportive comments and taking the time to review the manuscript.

Specific comments: Section 2.4: It is not entirely clear how the data treatment is done. Were the data time-shifted before submitting these to the eddy flux routine or did the eddy flux routine take account this shift and perform a covariance analysis? A better approach to deal with lag-time estimation when individual 30 min covariance functions are below the LOD (to estimate a lag time) is to average quality filtered covariance functions. This allows to obtain a study (or weekly) average lag time for each individual species. This is a more accurate approach if compounds exhibit different absorption and desorption properties along an inlet line.

We calculated the lag times for  $NO_x$  and CO data for each 30 minute period, by cross-covariance maximisation between pollutant concentration and vertical wind speed. When determining the lag time for each species a high-pass filter (Hartmann et al., 2018) is used which improves the precision of the determined lag time by an order of magnitude. As recommended, we then further calculated the median lag times for each species based only on high quality fluxes for each campaign. We performed the lag-time shift using these median values for each campaign.

We added more details to section 2.4 of the manuscript to make the data treatment clearer and have included the standard deviation when reporting median lag times.

Concentration data were coupled with wind data reported by the sonic anemometer by sub-sampling the wind vector data to match the 5 Hz concentration data. Data were then despiked prior to flux calculation as per the method described in Brock (1986) and Starkenburg et al. (2016). Following despiking, the lag time between vertical wind velocity measured in-situ on the tower and the pollutant concentrations, measured on the ground, was calculated. The lag time correction was determined by maximisation of the cross-covariance between pollutant concentration and the vertical wind component. When determining the lag time for each species a high-pass filter (Hartmann et al., 2018) was used which improves the precision of the determined lag time by an order of magnitude. The median lag time was then calculated for each species during each campaign. The lag time between the concentration and vertical wind speed during the winter campaign was found to be  $9.6 \pm 0.4$  s,  $10.0 \pm$ 0.4 s,  $10.2 \pm 0.3$  s for NO, NO2 and CO respectively. For the summer campaign lag times were calculated as  $9.4 \pm 0.4$  s,  $9.8 \pm 0.3$  s and  $10.6 \pm 0.5$  s. Because there was no discernible pattern or trend in the lagtimes and to prevent the flux bias that cross-covariance maximisation can introduce when fluxes are small (Langford et al., 2015), the final fluxes were calculated by applying the median time-lag value for each campaign to all flux periods.

Section 2.5: Line 6, 16: The authors should outline the changes incorporated in eddy4R that were specifically implemented and necessary for the present study (e.g. by showing a work flow diagram). Flux footprint model. The flux footprint model is based on Kljun et al., 2004 and was modified

according to Metzger et al. 2012. Kljun et al. 2012 updated their original footprint model. It is not clear what the exact differences between the cross-integrated footprints between Kljun et al. 2012 and Metzger et al. 2012 are, assuming they are not the same. A clarification of this issue would be warranted.

A workflow diagram (shown below) has been added to the manuscript which highlights the key stages in the flux processing workflow, with any settings specific to the Beijing campaign processing noted.